# Transitioning to Smart Cities in Gulf Cooperation Council Countries: The Role of Leadership and Organisational Culture

Ibrahim Mutambik [1,*], John Lee [2], Abdullah Almuqrin [1] and Justin Zuopeng Zhang [3]

1 Department of Information Science, College of Humanities and Social Sciences, King Saud University, Riyadh P.O. Box 11451, Saudi Arabia; aalmogren@ksu.edu.sa
2 School of Informatics, The University of Edinburgh, 10 Crichton St., Edinburgh EH8 9AB, UK; john.lee@ed.ac.uk
3 Department of Management, Coggin College of Business, University of North Florida, 1 UNF Drive, Building 42, Jacksonville, FL 32224, USA; justin.zhang@unf.edu
* Correspondence: imutambik@ksu.edu.sa

**Abstract:** The concept of Society 5.0, first introduced by Japan in 2016, has become a widely accepted model for the development of social infrastructures across the world. It is a model which is expected to take root globally over the next few years. It is also a model which has smart cities, which are connected and inclusive, at its core. The role of open data is critical to smart cities, and the ability to design and implement strategies for its use is a crucial element in their growth and success. This requires a leadership and organisational culture that embraces the concept of open government data (OGD) and understands its key role in the development of smart cities. In this paper, we examine how the leadership and organisational culture in Gulf Cooperation Council (GCC) Countries has impacted the progress of OGD initiatives and, therefore, the transition to smart cities. This is approached via a re-analysis of data from an earlier study in which semi-structured interviews were used to understand the views and attitudes of a range of senior government department personnel in OGD-related roles, where here a new thematic analysis seeks to identify clearer pointers to attitudes and practices directly relating to smart cities and Society 5.0. The focus on internal factors, such as leadership attitudes and organisational culture, as opposed to external factors, such as technology and resources, differentiates this research from previous studies and adds to our current knowledge. The findings lead to a discussion that identifies a likely gap in the leadership provided by more senior figures. A pilot study of a group of these leaders suggests a generalised problem with communication of policy, objectives and strategies, which is crucial to overcoming cultural impediments to smart city development. While further research is required, a need clearly emerges for significant changes in attitude and application at senior managerial and leadership levels if strategic goals are to be achieved. The paper also makes a number of specific recommendations for activities that could improve progress and indicate areas where more research would be beneficial.

**Keywords:** smart cities; open government data; e-government; Society 5.0; GCC countries

## 1. Introduction

Society 5.0 envisages a "super smart society" that uses the technological developments of the current information society (Society 4.0) to build a new, human-centred model of social infrastructure which provides 'new value and services' [1–4]. Such an infrastructure will depend heavily on the use of the related idea of the smart city–data-driven municipalities that use information and communication technologies (ICT) to increase operational efficiency and improve both the quality of government services and citizen welfare. The social and economic returns of smart cities are such that the concept is being realised across the world; the global smart cities market size is projected to be over USD 6 trillion by 2030, representing a compound annual growth rate of 25.2% from 2021 [5].

To optimise their efficiency and benefits, which include social, economic and environmental value [4,6], smart cities need to be built around infrastructures that encourage and allow for the free interchange of non-sensitive data by all public and private organisations. In fact, the need for governments to share information as part of smart city realisation is critical. Information sharing not only contributes to effective decision-making based on data but also provides citizens with the means to monitor and evaluate government performance in the management of services and welfare. This significantly affects the levels of trust and confidence that citizens have in their government [7]. In short, open (government) data (OGD) are critical to the delivery of urban management at all levels and the improvement of adaptability, flexibility, transparency, and response effectiveness [8].

While the openness of government and information is commonly seen as key to the development of smart cities, it is also widely recognised and studied as a major enabling factor for information-based societies in general. OGD has very broad relevance; it has, however, a special centrality for the smart city concept. In saying this, we do not wish to obscure the ambiguity, pointed out by Mutambik et al. [9], between "open government data" as "data that arises from the operation of open government" and as "government data that happens to have been made open". The latter may be limited, subject to censorship, etc., and not indicative of open government. Smart cities need to maximise the availability of data (and not only government data) to achieve their economic potential. However, it is arguable that smart cities will not achieve their true potential for the wellbeing of citizens, and specifically Society 5.0, without maximally open government data, as an aspect of truly open and equitable governance. This is discussed further in Section 5.

The issue of city development and management is relevant across the world. However, it is particularly relevant to GCC and Middle Eastern countries, as, by 2050, more than 90% of their aggregate population is expected to live in cities [10]. The implementation of a smart city philosophy is, therefore, of increasing importance for the region. Most of these countries have already formed a strategic plan for developing smart cities specifically [11], with developments underway, including at least the Masdar and Zayed smart city projects in Abu Dhabi, and NEOM, Amaala, Qiddiya and the Red Sea Project in Saudi Arabia [12]. Some GCC countries, such as the United Arab Emirates and Saudi Arabia, have introduced OGD platforms as an enabling phase. However, the performance of these platforms had not (as of March 2022) met expectations [13,14].

The GCC Countries are not alone in this failure to meet OGD objectives within a given timeframe. A recent analysis of the OGD portals of 60 countries actively moving towards Society 5.0, for example, found that most lacked basic usability features, and all had room for significant improvement [15]. Another study [7] of 34 smart cities (across 22 smart cities) of OGD portals also found that all had room for improvement in terms of the key metric of transparency.

These 'failings' could be for a number of reasons. One such reason, for example, is the complexity of the data ecosystem, which consists of many stakeholders and data channels, and requires considerable physical and financial resources to manage effectively. Another possible reason is based on culture and attitude: ideological and political barriers may exist at a leadership and management level, and these barriers must be overcome before OGD (and, by extension, smart cities and Society 5.0 principles) can be effectively implemented [16]. Where this is the case, there is evidence that there is significant resistance to radical change [17].

In this research, we seek to understand more deeply the lack of progress of GCC Countries in the development and implementation of OGD initiatives. We look specifically at organizational culture and ideology inside government departments to answer the following research questions:

1. To what extent are these aspects aligned with the objectives implicit in delivering Society 5.0?
2. What factors are inhibiting this alignment?

In an earlier investigation, Mutambik et al. [9] investigated progress on OGD in GCC countries through a focus on departmental managers and similar staff in public service tasked with implementing the developments. Semi-structured interviews were carried out with 24 participants (senior civil servants), and the results analysed using thematic analysis of attitudinal and behavioural themes concerning OGD and its deployment in general. For the purposes of the present study, the qualitative data arising from that investigation was re-analysed to seek more specifically insights concerning leadership, organisational culture and ideology and to examine these from the perspective of transitioning towards Smart Cities. Reanalysis of qualitative data is an approach that has been recognised for its capacity to generate valuable practical insights by extending or shifting the focus of analysis, e.g., [18–22]. In thematic analysis, for example, the identification and understanding of themes in data such as interview transcripts depend in part on the researcher seeing them as relevant to the investigation. While the original focus was on OGD and its deployment, the data were not interrogated specifically to seek insights into the organisational context and the role of leadership.

The re-analysis, reported below, revealed directions for further investigation and led to the implementation of a follow-up study of more senior leaders in the area. We discuss this below and argue that it helps us to articulate further questions about how progress in areas such as Smart Cities and Society 5.0 depends on complex aspects of the political and social structures in the region that are not yet clearly understood.

## 2. Literature Review

Open data can deliver a number of economic, technological and social benefits and are considered a key factor in the transformation of a 'conventional' city into a smart city [23–26]. This is particularly true when a country seeks to transition its social and governmental infrastructure to Society 5.0, i.e., to shift from an information society (Society 4.0) to a super smart society. Ultimately, open data are the driving force behind the development of a human-centred society that resolves social and economic challenges using the technological breakthroughs of Society 4.0 [4,6,27]

This, of course, raises the question of what 'open government data' actually is. How, in other words, is it defined? In fact, there are many definitions, most of which broadly agree. The OECD, for example, defines OGD as 'a set of policies that promotes transparency, accountability and value creation by making government data available to all' [28,29], while the Austrian government defines it as 'databases that contain non-personal and non-infrastructure-critical data that are freely accessible for public use' [29]. However, while such definitions agree in terms of the general accessibility and usability of data, they do not provide specific technical conditions that data must fulfil in order to qualify as 'open'. These conditions are that, for data to be 'open', it should meet eight principal conditions: i.e., it should be primary, comprehensive, accessible, timely, machine-readable, non-discriminatory non-proprietary, and without charge [13,14,30].

As already noted, the number of smart city development projects across the world is growing rapidly. As open data are a prerequisite for the transition to a smart city, it would be expected that the implementation of OGD projects would also increase. The statistics support this expectation. The OECD's Open Government Data Project reports on the progress of over 30 countries in the use of OGD [31–33], and many other countries have integrated OGD strategies into their political, social and economic agendas [34].

However, while there are numerous benefits to using OGD, there are also risks and problems involved with its adoption. One of these difficulties, for example, is addressing the concerns of leaders that the use of OGD does not deliver measurable or significant benefits [35]. Others have expressed fears that their business interests might be negatively impacted by the release of commercially sensitive data, in the same way, that software developers can be reluctant to participate in the open-source sector as it requires the release of unique and proprietary code [36]. This can affect licensing revenues. Another concern of leaders is that the implementation of OGD could add an unacceptable workload to employees or that the department or enterprise concerned lacks the time or financial

resources. This can be a major issue, as managing datasets (creation, updating, etc.) can involve significant costs.

Lack of resources is not the only factor that can be an obstacle to the implementation of an OGD initiative. There is also the challenge of privacy protection. It is the responsibility of data providers to ensure that no personal or sensitive information is published [37]. The process of ensuring this can be complex and can carry the risk of personal and/or corporate prosecution. A similar situation applies to the issue of data ownership: it is often unclear who ultimately owns the relevant data, and publication can therefore result in a licence or privacy violation [38]. The result can be prosecution. This particular concern (data ownership) is frequently an issue within government departments in GCC Countries, as data sharing is not, historically, part of the systemic structure [35], so data ownership is often very unclear. This can lead to a reluctance to progress OGD initiatives.

As well as the challenges of process and resources described above, open data initiatives can also be inhibited by technical issues. These include a lack of standardisation of data formats [36], no uniformity in format conversion or output [38,39], and a lack of standardisation of OGD metadata [38], despite the existence of general metadata standards, such as the Dublin Core [40].

All of these risks and challenges, either individually or collectively, can impede the progress of OGD initiatives, as eliminating or mitigating them usually requires significant investment. The issue is further compounded by the fact that promoting and encouraging the adoption of OGD at scale requires the provision of guidance and user support. This involves further cost. However, it depends crucially on the leadership and motivation shown by those responsible for driving the necessary changes and developments. In GCC countries, in particular, it appears that unless leaders and managers can be convinced that OGD initiatives are low-risk and will deliver value for money and significant Return on Investment, the implementation of such initiatives will not progress as rapidly as many hope [9], and this may impose a serious impediment to the transition to smart cities and Society 5.0.

### 2.1. International Comparisons

All of the challenges described above have been met and, with varying degrees of success, overcome in a number of countries. This is evidenced by the countries with relatively high marks on the OECD OURdata index [28], the "Open Data Barometer" [41] and the Global Data Barometer [42]. It is notable that the GCC Countries either show a low score on these indices [43] or are absent altogether. In this study, we investigate the extent to which the reasons for this are embedded in the culture and mindset of departmental leadership and consider factors that may be hindering the rate of transition to smart cities.

In other international contexts, similar questions have been investigated. For example, in China, Zhang et al. [44] have examined the factors whereby officials implementing OGD initiatives are incentivised or otherwise, advocating especially the need to alleviate risks to the individual so that "any adverse consequences should not be held accountable to the staff". Ruijer et al. [45] study examples in the Netherlands and France, noting that several issues may combine, including organisational inertia and political preferences that can produce "strategically opaque transparency". Hossain et al. [46] observe that "fostering [government] agency participation in OGD initiatives is a critical concern"; investigating the example of Australia, they find that "lack of political commitment, and external pressure" are among significant hindering factors. Garcia [47] finds in the Azores that a "[s]ilo mentality from a closed organizational culture is perceived as the main obstacle to the OGD initiative". In the GCC, we find factors similar to all of these, but we also note the influence of a cultural context that is distinctively different from any of them, as we discuss in detail later.

Authoritarianism

The political systems in the GCC countries are among those that are often characterised as "authoritarian" [48,49]. Some researchers suggest that there is a tension between the nature of such systems and the "openness" built into the concept of OGD. They observe that the idea of a Smart City, in which the collection and use of data is ubiquitous, may lead to an enhanced potential for censorship, surveillance and control [50,51], running counter to the idealistic narratives of open government and citizen empowerment. More broadly, it has been contended that smart city developments in the global south and east are often very different from those in the north and west and that some of those in the Middle East may have more in common with the former [48]. In the present study, we have not addressed these issues directly, in part because they typically represent external perspectives, whereas we are trying to get a clearer view of how smart city developments in the GCC are seen by those working to implement them. However, we return to this point in the discussion.

## 3. Research Method

### 3.1. Investigation Aims and Structure

The aim of this research is to gain insights into the lack of progress of GCC Countries in transitioning to smart cities. As acceptance and implementation of open data are key elements in the transition process, the attitudes of departmental leadership and management towards OGD, as well as organisational culture, are likely to provide important indicators. In an earlier investigation [9], the opinions and perspectives of 24 senior personnel towards both the theory and practice of OGD were sought using semi-structured interviews, and the data analysed to examine progress on the implementation of OGD policies. The present investigation sought to re-analyse those data with a view to gaining deeper insights into the roles of organisational culture and leadership in transitioning towards Smart Cities.

### 3.2. Main Features of the Previous Study

The sample for the study consisted of 24 participants, selected using a combination of snowball and purposive sampling, from different GCC Countries and government departments. The sample was limited to 24, as there was clear evidence that this was the saturation point [52–54]: the "saturation method" refers to a process used to determine when data collection is complete—the point at which collecting additional data no longer provides new or useful information. During the data collection and analysis, the researchers continually assessed whether new data were adding to the understanding of the research topic. After, 24 of the researchers determined that additional data were no longer providing new insights with the last three interviews, and therefore, data collection was considered to be saturated and sufficient for the current study.

All participants had significant experience in the implementation and management of OGD initiatives across a range of data types. Table 1 shows the job area and experience and the state of origin within the GCC countries of each participant. There is broad representation from across these countries.

The interviews were relatively unstructured, based on cue questions that allowed a broader discussion to develop. Specific topics addressed included:

- The nature and benefits of smart cities and the role/importance of OGD.
- Internal attitudes towards the policy goal (smart city transition) could hinder progress by a government department in implementing OGD initiatives.
- Aspects of organisational culture that could obstruct the progress of an OGD initiative.
- Perceived obstacles, either internal or external, to the implementation of an OGD initiative.
- The result(s) of lack of progress towards OGD.

**Table 1.** Summary of details of interviewees.

| Participant | Role | Years of Experience | Country |
| --- | --- | --- | --- |
| P1 | Government official, open data initiative | 16 | UAE |
| P2 | City planner | 6 | Bahrain |
| P3 | Data scientist/analyst | 11 | UAE |
| P4 | Urban Planner | 4 | Bahrain |
| P5 | Open Data Coordinator | 13 | Oman |
| P6 | Geographic Information System Analyst | 9 | Bahrain |
| P7 | Transportation Engineer | 7 | Saudi |
| P8 | Smart Grid Manager | 6 | Oman |
| P9 | Smart City Manager | 8 | UAE |
| P10 | Open Data Coordinator | 7 | Kuwait |
| P11 | City planner | 6 | Saudi |
| P12 | City planner | 5 | UAE |
| P13 | Open Data Coordinator | 3 | Qatar |
| P14 | Urban Planner | 3 | Kuwait |
| P15 | Open Data Coordinator | 4 | Saudi |
| P16 | Smart City Manager | 14 | Oman |
| P17 | Geographic Information System Analyst | 7 | Oman |
| P18 | Transportation Engineer | 4 | Qatar |
| P19 | Smart Grid Manager | 10 | UAE |
| P20 | Smart City Manager | 12 | Saudi |
| P21 | Open Data Officer | 2 | Qatar |
| P22 | Open Data Coordinator | 7 | Kuwait |
| P23 | Smart Grid Manager | 5 | Qatar |
| P24 | City planner | 14 | Saudi |

A verbatim transcription was made immediately after each interview. To help ensure accuracy and consistency, each recording was matched against the field notes by two separate and independent researchers. All interviews lasted approximately the same time (one hour) and were carried out, transcribed verbatim and analysed in the main language of the interviews—i.e., Arabic. The information was merely translated into English for reporting purposes; hence, there can be minor differences between translations of quotations from the data in the previous study and in the present one. It was established, for the purposes of research ethics, that the re-analysis of the data described here would not infringe on the agreement made with the participants about the uses of the data.

*3.3. Data Re-Analysis*

The interview transcripts were re-analysed using a thematic analysis approach consistent with standard guidelines [55,56]. Thematic analysis was chosen as it is a flexible and effective method of collecting and examining the views and perspectives of different participants, allowing the identification of similarities and differences and generating insights [55,57]. The method was considered particularly appropriate for developing a better understanding of the issues covered by the current research questions, that is, to understand the source(s) and reason(s) for the misalignment between departmental culture or ideology and the objectives of Society 5.0 and, hence, the lack of OGD progress in GCC

Countries. It is noted that although thematic analysis is flexible and powerful, it can also produce inconsistency and a lack of coherence [58–60]. While care was taken to avoid these issues, it should be considered a possible limitation of the study.

### 3.4. Coding and Producing the Analysis

The (re-)coding of each transcript followed a process based on that described by Braun and Clarke [61] and included elements of Grounded Theory as described by Charmaz [62]. The steps were as follows:

- Initial coding. This employed a segment-by-segment coding technique, allowing the identification of similarities and differences in interview content. This stage also resulted in the emergence of patterns and connections to the research questions.
- Focused coding. These are codes that are considered to be of particular significance [63]. Once found, they were grouped to build patterns within each interview, resulting in a data summary that did not alter the initial significance of the interviewee's comments. The researcher also utilized focused coding to 'reveal his own biases' about the study issue [62].
- Theme search. A 'theme' is defined as "a recurring regularity emerging from the analysis of qualitative data" [64], p. 470. As a first step in this phase, the focused codes were recorded on a worksheet to allow the identification of groups that shared ideas or attributes of meaning. These common ideas are known as 'sub-themes'). These were then analysed for shared characteristics at an advanced level in order to form 'main themes'. These were then grouped into 'major themes' relating to the research question.
- Theme identification. This was carried out in accordance with the recommendations of Patton [65]. These involve the use of dual criteria for creating themes: (a) internal homogeneity (meaningful coherence within themes) and (b) external homogeneity (a clear and identifiable distinction between themes).

## 4. Results

The coding analysis identified twelve sub-themes and six main themes (Table 2). These main themes and sub-themes are shown in detail below, together with examples of text taken from interviews that formed the basis of the study analysis.

**Table 2.** The finding of analysis (main themes and sub-themes).

| Main Themes | Sub-Themes |
| --- | --- |
| Ultimate purpose/aim | Operational confusion<br>Lack of public acceptance |
| Planning and consensus | Lack of clear strategy<br>Lack of internal consensus |
| Risk | Societal and governmental<br>Personal |
| Leadership and resources | Human resources<br>Leadership |
| Information | Data security and value<br>Data compliance |
| Perceived benefits | Government<br>Public |

### 4.1. Ultimate Purpose

A lack of understanding of how a smart city will operate in detail and a lack of conviction concerning its benefits emerged as a clear theme.

Operational Confusion and Lack of Public Acceptance

These follow quite closely some of the observations of the original analysis of the data in finding a clear recognition among participants, with remarks such as (R17):

*Smart cities sound a good idea on paper, but there is considerable confusion regarding how specific OGD initiatives can integrate to make them work in reality.*

Sometimes there was evidence of a very clear understanding of how public acceptance is a key factor in smart city initiatives, yet it is seriously inadequate:

*A major issue is that many people are unaware of the term 'smart city' and its implications, and have little idea how to use open data. With this lack of understanding and acceptance, OGD initiatives are likely to provide poor returns, so they are not being implemented effectively. Matters are further complicated by the fact that cultural differences mean that public confusion differs in type and strength from country to country, so it's hard to standardise an implementation strategy. (R19)*

This point was supported by other government departments, who reported that very few companies or individual members of the public had made requests for services based on open data.

*Demand is a key driver of development of any product or service. But this is almost entirely absent from the public and commercial enterprises just now. This may be due to a lack of awareness of the benefits of [OGD], but whatever the reason, data providers not be motivated to act. I imagine this is the case across all or most countries in the region. (R4)*

There is also a recognition that developing demand will depend on public education and information. As R11 phrased it:

*There have been publicity campaigns by GCC Countries to try and encourage understanding and engagement, but they didn't have much effect. This is concerning, as unless the public realise how open data can apply to their problems and help them find smart solutions, adoption is likely to remain low. This, in turn, will mean government departments will lack motivation to implement OGD. (R11)*

It hence emerges that the ultimate purpose of smart cities is unclear both to the public who will inhabit them or otherwise stand to benefit from their development and to those whose role is to facilitate the development. This suggests a lack of focus on the issue in education and in informational processes across society. It may also indicate, as we suggest in Section 5.3, a kind of contradiction between certain cultural factors that stands in the way of clearly articulating the purpose and cultural implications of smart cities.

*4.2. Planning and Consensus*

Becoming a smart city does not happen by itself. It is a process that requires long-term planning and strategy by the governing body, as well as support from external bodies such as the general public and private organisations. The current research showed that both of these factors were either missing or very weak.

4.2.1. Lack of a Clear Strategy

A perceived lack of strategic planning at the higher levels of government emerged as a notable factor. This applied to the ultimate goal of meeting the criteria for being a smart city, as well as the medium-term objectives of implementing OGD initiatives. For example:

*Medium-term open data policies, and their implementation, will usually be a function of longer-term smart city objectives. This means it's essential to understand 'where we are going' in terms of long-term goals. This vision seems to be missing at a government level–or, if it isn't, then it hasn't been widely publicised. Until departments know the long-term strategic goals, they are unlikely to be enthusiastic about using open data. (R3)*

The lack of a clear strategic direction is not only reflected in the reluctance to design and implement OGD policy but also shows in more practical aspects of departmental

management, such as the recruitment of appropriately qualified (OGD-experienced) staff. The effectiveness of engagement is often a result of the personal beliefs and vision of individual leaders:

> *It depends on the level of commitment of senior personnel, and their alignment with the philosophy of smart city design . . . if, for example, a data provider has a leader who is personally in tune with the smart city, then they're more likely to push strategy development, both within and beyond their department. However, there's not much evidence of this happening. (R1)*

> *Lack of strategic leadership can often result in a lack of initiative at a departmental level. Without proper direction, data providers generally have to make an individual decision to develop and implement OGD processes and initiatives. They're often unwilling to do this, for various reasons, such as concerns about financial and accountability issues. (R17)*

### 4.2.2. Lack of Internal Consensus

Transitioning to a smart city is a major project which requires clear strategic direction. Developing and implementing such a strategy demands a shared vision and consensus at all senior levels. As has been noted, there is evidence from this study that strategic direction is unclear. Several participants also remarked on the lack of required consensus among senior leaders and government departments on desirable outcomes. This may be a contributor to the lack of strategic direction. As R24 phrased it:

> *The development of strategy needs clear agreement on defined endpoints and outcomes– social, economic and political. As things stand, there seems to be no such agreement, so there are few starting points for departments to frame objectives for OGD. As a result, most data providers have not developed OGD initiatives. (R24)*

Inevitably, a lack of consensus can mean that objectives can change mid-project, which can be a significant waste of time and resources. For example:

> *Even when there is economic or social merit in an OGD project, we are often unwilling to implement it, unless there is cross-departmental agreement and support at the highest levels. Without this, there is no certainty that objectives won't be redefined during or after implementation. (R22)*

Here, we see overall not merely the lack of a clear strategy but the existence of an environment that works strongly against the capacity to develop or implement a strategy. A significant shift in the priorities of the prevailing culture will be needed to address this. The evident need for "support at the highest levels" points to a need for the government itself to provide and articulate a very strong focus and steer that could re-orient the approach of departments at all levels. We return to this point in Section 5.3.

### 4.3. Risk

Perceptions of risk emerged as an especially salient area of concern in the previous analysis and remained a very clear issue in the re-analysis. We classify risk under two principal headings: risk to government and society as well as risk to individuals.

### 4.3.1. Risk to Government and Society

Risks to the government can include, for example, financial loss, as well as the possibility of losing some control over specific aspects of government. The risks to society can also take several forms, from poor delivery of services to a failure to provide commercial opportunities. For instance, one participant explained the following:

> *The fact is that, by their nature, government departments act to minimise risk. This means that if there's any level of doubt as to whether data should be released, a department will not do so, as it's almost always safer not to provide data than to provide it under such conditions. (R20)*

Although not always openly acknowledged, a key concern for many GCC governments is maintaining control and power. This can be compromised by the release of data (R21, R22). As a result, many departments are hesitant to embrace OGD. This was highlighted by one participant, who asserted the following:

> *By releasing raw data . . . we take the risk that it [the data] could result in the exposure of any errors, shortcomings or other flaws in our processes. (R22)*

As well as political risk, OGD may be seen as involving financial risk. Data may represent a source of revenue, and by releasing data, this revenue source could be compromised. According to R8:

> *Data is key to helping many government organisations in GCC Countries meet their performance targets. To release open data could significantly reduce their income, so they are reluctant to do so. (R8)*

There are also societal risks associated with OGD. According to R4:

> *We often focus too much on the risk of OGD to government, and lose sight of the fact that it can represent a risk for society, too. A failed, or underperforming, initiative, for example, can result in financial losses to a department, which means it has no choice but to reduce services, or quality of service. In extreme circumstances, it can result in increases in central taxes. There are many scenarios where OGD failure or underperformance can negatively impact social welfare or infrastructure.*

### 4.3.2. Personal Risk

While the risks of implementing OGD can be significant at the government level, they can be equally significant at a personal level. Many participants either implied or explicitly stated that the progress of OGD initiatives can be impeded by an individual desire to avoid accountability and responsibility, as the risks can be high. According to participant R10:

> *Open data initiatives can be complex, and there's no certainty that they will go to plan. If anything goes wrong, you can find yourself in a difficult position. For most people in leadership roles, the rewards aren't high enough to justify taking risks, so they end up doing nothing. (R10)*

Other participants shared this view. For example:

> *I can appreciate that there are benefits of open data to government, and to society, but the risks at a personal level are quite high. Usually, I go for a compromise solution, such as releasing statistics on the organisation–it has much more predictable consequences, and is usually just as acceptable to users. (R7)*

The nature of this issue of risk seems to highlight the systemic nature of the difficulties faced in this area by the public sector in GCC countries. A framework needs to exist in which these kinds of risks are not, in general, borne by individual departments. If there were a basic presumption in favour of publishing data, then the risk would be more in failing to publish when required. Perhaps, also, the categories of data that should not be published are currently drawn too widely: if they were limited mainly to personal data of the kind protected by the European Union's GDPR, for instance, then in most cases, the risk of publishing such data could inadvertently be kept quite low.

### 4.4. Leadership and Resources

As with commercial operations, government departments are under great pressure to optimise the use of resources. This can have a significant impact on the ability and willingness of leaders to implement initiatives of all kinds, especially those connected with OGD, which can be complex and expensive to design and implement. It can also inhibit individuals from taking on leadership roles, as the extra workload, pressure and personal risk are often not reflected in the reward. It was clear from the results of the analysis of interviews that the lack of suitably qualified staff and the lack of willingness to take on

leadership roles are two of the major factors that impede the progress of OGD initiatives. This is distinct from the issue, which underlies many of the observations reported here, of how effectively leaders, when they are in place, understand and support the objectives and methods required for OGD initiatives to progress. We address this more specifically in the Discussion section.

### 4.4.1. Human Resources

The process of OGD implementation is not only complex and time-consuming, but requires qualified and experienced staff. This, according to several interviewees, can represent a major issue for many departments that lack such staff. As one participant said:

*The work is very intensive, requiring a considerable amount of overtime, so we often struggle to meet deadlines, or even finish a task at all. The only solution is to hire more staff, but the level of qualification we need means that such staff are hard to find–they are often attracted by other positions which pay higher rates. (R9)*

This problem with staff recruitment was mentioned by many other interviewees. According to participant R15, for example:

*There is very little support for recruiting suitable staff, so many employees end up taking on dual roles, adding extra pressure. This can be a false economy, as it can lead to errors which have to be fixed, taking up even more time. This is one of the main reasons that our projects often run behind schedule. (R15)*

A similar point was made by R12.

*We have looked closely at the OGD issue, but simply don't have the human or financial resources to implement it. The main problem is people-suitably qualified staff are difficult to source, and expensive as well. (R12)*

### 4.4.2. Leadership

As OGD initiatives are workload-intensive, carry high risk, and often offer no extra reward, many senior personnel are unwilling to assume roles that involve leading OGD projects. Several participants highlighted this point. According to R1, for example:

*I know from colleagues who do it, that OGD initiatives involve considerable extra work, which makes me think twice about taking on a leadership role. Although I can see some benefits to implementing OGD, I would be forced to compromise in the other areas of my work, which is something I don't want to do. (R1)*

Or R9:

*Most senior departmental personnel are already under high pressure–they certainly don't need the extra pressure of leading an OGD project, especially when the position offers no significant benefit to either the person concerned, or their department. (R9)*

### 4.5. Information

Progress on OGD is also dependent on the integrity of the data that may be made openly available. We identify concerns in this area as falling under two sub-themes:

### 4.5.1. Data Security and Value

In terms of security, most departments will only release data that are known to be fully safe, which tends to restrict 'qualifying' data considerably. A similar situation applies to value; a significant amount of data is considered to have commercial value and is, therefore, not released. Data that needs processing prior to release (adding to time and cost) is also held back. The result is that most departments will only release data of poor quality. As one interviewee put it:

*I'll only release data if I'm sure it's completely risk-free, which is rarely the case. It can be very hard to predict how raw data will be used, so it's usually safer to hold it back. (R5)*

A number of interviewees stressed how the value of data could be reduced, or completely negated, by the release decision-making process. For example (R2):

*The commercial value of data is usually related to its age. Old data is worthless. For it to be valuable, data needs to be real-time, or near real-time. However, we sometimes get data that has taken so long to check that it's useless by the time it's released. It's no surprise that commercial interest in open data isn't particularly high or growing. (R2)*

### 4.5.2. Data Compliance

Another concern is legal compliance. It was noted by several participants that the law surrounding data disclosure could be complex, and departments often claim that they are legally prevented from releasing data. One interviewee remarked that:

*We tend to take no risks. Unless we're completely certain that it's safe to release data, we refer it to the appropriate court for a decision. Only if the court gives it the go-ahead will be consider making it publicly available. (R3)*

Another participant made a similar point:

*It's a complex legal framework, and you could easily find yourself being non-compliant if you release data without the court's viewpoint. This can take time and money, so data is often not released, rather than risk legal issues.*

Again, we see here that the problem in the government departments reflects something which is actually much wider. Even if departmental leaderships become much more embracing of the goals of OGD, they will remain subject to constraints that make it impossible to deliver properly. Although it will, of course, not be a "quick fix", there is a need for a major overhaul of legal frameworks and top-level policies around key issues of what data are for and how they are used, and ultimately, the role and nature of public services in GCC countries.

### 4.6. Perceived Benefits

If OGD initiatives and policies are to be embraced and enacted by governments, they must offer clear and significant benefits. In the context of the move towards Society 5.0, through the transition to smart cities, OGD is believed by many to be able to deliver benefits both to governing bodies as well as society in a more general sense. However, a significant number of participants in this research felt that there are no clear benefits of OGD, either to governments or the public. Many expressed the view that OGD is, essentially, a waste of valuable resources, imposing a major financial burden on the relevant department while delivering little, if any, return on investment at any level. This signal, very clear in the original analysis of the data, is persistent.

### 4.6.1. Government

Many proponents of smart cities argue that governing bodies can benefit from 'smart government', which offers advantages such as more cost-effective decision-making, better planning and more efficient management and control. This, according to the current research, is not the case in GCC Countries. There is a problem that any wider economic benefits, even if recognised at all, are not seen as emerging at the local departmental level. One participant put it bluntly:

*Personally, I can't see any obvious benefits [of OGD] to government departments. It's true that we sometimes hear talk of improved GDP, but I fail to see how this will help us at a departmental level. Without a clear argument that OGD will improve departmental performance, there's no incentive to provide it. (R7)*

Other participants made a similar point: OGD might deliver benefits to others, but it is unlikely to deliver benefits at a departmental level. Thus:

> *It seems to me that there's a real danger that open data would work in favour of external parties, such as other countries, without helping the public or the government departments of the country which provided the data. (R4)*

### 4.6.2. Public

Proponents of smart cities also argue that the concept also benefits the public by (for instance) improving services, improving the quality of living, offering faster and better solutions to individual problems, and improving commercial opportunities. Again, the participants in this study did not agree. As one participant (R11) phrased the issue:

> *Delivering open data will simply increase costs without improving services. Over time, this can only have a negative impact on the government's reputation in the eyes of the public. This makes the whole idea pointless. (R11)*

Other participants echoed this opinion. For example:

> *OGD is supposed to be useful to society, but I personally find it hard to see how, and I think most of the public feel the same. Of course, this could be because OGD hasn't been around very long, and there's low levels of awareness and understanding among both the public and data providers. But unless something is done about this quite quickly, few departments will feel justified in starting an OGD initiative.*

## 5. Discussion

### 5.1. Challenges

The challenges faced by governments and governmental departments in developing and implementing OGD initiatives have been the subject of several previous studies [38,66,67]. Most of these studies have reported on factors that impede the progress of such initiatives, and all of them have contributed important insights. However, they have also focused on external factors which create barriers to OGD progress, and often not internal factors such as organisational culture and effective leadership. These factors have always been important, but they have become more so in recent years, as open data are key enablers in the transition to smart government, smart cities and Society 5.0. Unless organisational culture and the abilities of leadership are aligned with the vision of a smart world, they are unlikely to provide the necessary drive that is required for successful OGD implementation. It is in this respect that this research contributes to existing knowledge. We are often observing factors and effects that have been observed in previous studies, usually in different cultural contexts: we embrace the commonalities between these and our findings, but we focus on the distinctive aspects of the context of the GCC countries. The examination and analysis of the attitudinal, cultural and leadership aspects of OGD initiatives in GCC Countries, as presented in Section 4, provides insights as to whether these factors are hindering these countries in their journey to Society 5.0.

### 5.2. Perceptions of Risk

Our suggestion that management lacks ideological or political motivation for OGD implementation is illustrated by the findings of the study connected to risk. The importance of minimising risk is discussed in most studies of open data challenges [68–70], and some of these studies have noted the risks, at an organisational level, associated with overloading staff by increasing workloads [69,71], as well as the risk of compliance and legal issues. Few studies, however, have reported a perceived failure to deliver economic growth as a reason for poor OGD progress. This is worth noting, as GDP growth is frequently cited as a major motivator for OGD, together with political improvements resulting in enhanced public understanding and opinion of government actions [39]. The findings of the present study, however, indicate that the priority of data providers in GCC Countries is departmental gain (mainly financial) rather than national economic growth. The implications of this are clear: even if the idea of OGD has been recognised and embraced at a government level, the perspective has not yet been adopted at a departmental level.

Another aspect of risk that has not been widely studied in an OGD context is the personal risk borne by leaders and potential leaders of OGD projects. The findings of this study indicate that the risk of being held accountable for failures or underperformance is significant within GCC Countries. This seems to be preventing many able and qualified individuals from seeking or accepting leadership positions and is contributing to the lack of progress in OGD implementation. This concern appears to have been most clearly noted elsewhere in the context of China [44], but it may also be more widespread.

An important step that would aid the progress of OGD is improved education. It was clear from the results of this study that many leaders and senior personnel were either unclear about or unconvinced of the benefits of open data, both to their own department and to wider society [39,72]. This makes the risk-reward equation unfavourable to most departmental managers and therefore acts against the interests of OGD deployment. This implies that suitable education and communication programmes should be deployed using a range of internal, external or online delivery mechanisms [39,66,73]. Emphasis should be placed on specific internal benefits of open data, such as the fact that they can lead to improvements in departmental management and services, increased resources and favourable publicity. However, the wider societal benefits of smart government, such as economic growth and better public services, should not be ignored. Examples of how other countries are benefiting from OGD would be a powerful tool [74]. Building awareness and understanding of OGD benefits, in terms of both immediate departmental advantages and longer-term public benefits, is an important element in ensuring that key personnel become advocates for smart cities [75].

A further finding of the study was that even when government departments believe in the goal of an OGD initiative, they are often sceptical about whether or not it will progress successfully, often resulting in a decision not to proceed. To address this, open data, for specific purposes or as a general principle, should be integrated into national policy and legislation. In general, a variety of legislative frameworks and policies often encourage the development of smart cities and open data can be utilized to promote sustainable development, improve public safety, improve transportation and boost economic growth [76,77]. Many countries have passed legislation and regulations to protect people's privacy and data security in the context of smart city technologies [77], as well as to address environmental concerns and issues in digital infrastructure development. Overall, the legislative foundation for smart city development reflects a growing realization of the potential benefits of new technologies for improving urban living and resolving many of the difficulties that cities face today. This approach is likely to be effective within the GCC countries only if OGD authorities are delegated with greater powers to approve OGD projects within these legislative parameters.

OGD is still a relatively new arrival in GCC countries [13,66]. This is a significant factor in its lack of progress to date and explains why so many of the concerns of participants in the study were related to practical issues, such as understanding which data types are suitable for publication and questions about resources. One commonly-raised issue was the exact definition of 'open government data'. It was clear that there is a diversity of views on what constitutes OGD and that this lack of clarity and cross-departmental consistency can lead to a reluctance to progress with initiatives.

All of the technical and practical challenges to the progress of OGD, such as how to ensure its consistency and accuracy and how to resource it, are significant. However, there are also ideological questions that must be addressed if OGD is to be successful. The importance of one such question was made clear by the responses from the participants in this study. This is the question of whether the implications of OGD are ultimately desirable. While there may be practical benefits to OGD, many respondents also raised the issue of whether open data are inherently desirable.

We mentioned, in Section 1, the ambiguity of the phrase "Open Government Data", which can be interpreted as (a) data that are, by default, not freely available and is only made so by government decision, or on the other hand as (b) data that arises naturally from Open

Government, and hence is open by default, closed only by explicit government decision. The reality of the difference between these two definitions is well illustrated by the fact that none of the GCC countries is part of the OECD's Open Government Partnership [31,78,79], even though they claim to be committed to the concept of OGD. This implies that GCC Countries are ideologically aligned with approach (a) (data are closed by default). However, this approach also depends on the ongoing assessment of datasets to decide which can be made available and when. This is a complex, time-consuming, expensive and risky process, which the participants in this study found to be generally undesirable, a mindset that is at least in conflict with and may be contrary to, government policy. As, in the end, true OGD is inherently connected to the principles of open government, this is a conflict that will need to be resolved at the highest level of leadership if the goal of transitioning to smart cities and Society 5.0 is to be achieved.

Although resolving this conflict will not be easy, it has practical as well as ideological implications. One of these is the reduction of the perceived risk mentioned above among departmental leaders and management. While perceived risk among civil servants has been examined by several studies [66,71], the concept has not been clearly defined or described. The results of this research address this issue by showing that departmental leaders in the GCC countries perceive risk in two main ways. The first is the risk of being held accountable and criticised for releasing data; the second is the risk of losing vested interests in the form of the value attached to data, which is usually considered to be a strategic resource [71]. However, neither of these forms of perceived risk would arise under a truly open government, which, we argue, integrates criticism and accountability into its culture and removes dependence on vested interests. Of course, it is true that, in practice, these risks still exist, even within an environment of open government, but it is also true that they are significantly less influential in shaping the attitudes and actions of staff.

*5.3. The Bigger Picture*

Using thematic analysis, this re-analysis identified that the factors which negatively impact the progress on Smart City development in GCC Countries, represented especially by considerations around OGD, can be categorised under six main themes, as shown in Table 2: (a) Ultimate purpose/aim, (b) Planning and consensus, (c) Risk, (d) Leadership and resources, (e) Information, and (f) Perceived benefits. Although these categories typically characterise the definition and implementation of a political strategy, the responses within each theme and sub-theme of this study indicate that the views of departmental management personnel are not well-aligned with indicators associated with fast or successful OGD implementation and hence are not optimally promoting Smart City development. As we have seen above, considerations around risk, in particular, limit the capability of managers to engage with these developments.

Another important finding of this study, which is related to embedded cultural attitudes, is that management personnel do not feel that OGD represents a significant RoI (Return on Investment) for society as a whole. This may be a result of the fact that the concept is relatively new to GCC Countries and has therefore had insufficient time to 'prove itself'. This contention is supported by the fact that, in countries where open data is relatively well established, the social and economic value at a national level is clear [78]. However, it may also be because political and organisational cultures, which are inherently conservative, are more focused on an internal definition of 'value', i.e., they connect the concept of value with improvements in internal management and services and organisational capabilities. In either case, the study suggests that, at this level, the personnel are not ideologically or politically motivated to encourage OGD implementation.

One reason for this may be that they lack a view of the "bigger picture" of developments such as the concept of the Smart City, within which OGD may be seen as playing a role that transcends purely departmental concerns.

To develop the present discussion and as a preliminary investigation of this possibility, we carried out a series of further interviews of leaders at a more strategic level, seeking, in

particular, to reveal their thinking around the relationships between the idea of the Smart City and the uses of data. These semi-structured interviews comprised a group of five individuals. We did not subject the outcomes to a full thematic analysis because we were interested in identifying insights that could be related to the themes already found in the earlier interviews. The interviews were conducted in Arabic and transcribed and translated to English for the purposes of the present discussion. This might be seen as a pilot for a broader and more detailed study in the future.

A much more strategic level of understanding emerges quickly from these interviews. Participants recognise that Smart Cities is an objective that needs to be explicitly worked towards. Concerning, for example, the potential of open data, we find views such as:

Open data can be used to solve most problems, and to find smart mobility (transport and information and communications technology), the smart environment (natural resources), and smart life (quality of life). (PP1)

*For information on the availability of parking lots, garbage collection, and public lighting, communities can first look at their own systems. Other national government data can be used to promote this, including information on weather trends and even energy use. To obtain useful information across a wide range of categories, from traffic trends to air quality, sensors and IoT cameras can be included from private companies. (PP1)*

*As you begin to think about data products and services, every city official and service provider should note that, while it may be the end result, data services are not just about monetizing data. Consider analytics and information management, including platforms that can integrate data and automate data processes to enhance speed, accuracy, and data integrity, when developing data-driven services. (PP2)*

*Data interoperability and data management is critical to getting the most out of data from a range of sources and systems, including commercial and semi-private organizations, . . . City operators and providers must embrace open APIs and open data. (PP2)*

*Our primary focus is on data, collection and processing, we want to help our customers who are now mostly in the government sector, produce more value from their data by bringing it to life and developing new services based on it . . . [T]he growth of smart cities is closely linked to open data; so that open services and open data are increasingly pervasive in smart cities. (PP3)*

*[S]mart data processing is based on the principles of sharing and openness, which are just two examples of the "smart" frame of mind that an entire city must adopt in order to be considered a smart city. (PP3)*

In these interviews, we find no suggestion of the need for an individual department to justify a focus on open data or to demonstrate that it can cover any associated costs. Although risks such as privacy and security are recognised, there is no suggestion that the process of implementing strategies based on OGD will be risky for departments or individuals in public organisations [80]. In this sense, there appears to be something of a disconnect between the perceptions of these interviewees and those of the management personnel interviewed earlier. This could be seen as, on the one hand, a failure at the strategic level to appreciate the practical difficulties of implementation and, on the other hand, a failure of implementors to appreciate the wider value and motivation of the innovations they are being asked to implement [81,82].

We noted earlier that there is a scepticism among managers that the public are supportive of OGD initiatives. At the more strategic level, although there is a recognition that extensive public support will be needed, there seems to be a rather tacit expectation that it will arise, perhaps from the benefits of data-based services being sufficiently self-evident.

These observations indicate, we suggest, a general lack of communication, both vertically between the levels of the public services in terms of strategy and management and more horizontally between these services and the public. Managers do not seem to have recognised and understood the strategic message about the goals of Smart Cities

and the pivotal role of developments such as OGD in realising them, while, at the same time, thinking at the strategic level seems not to recognise the problems this is causing for implementation. Meanwhile, the public seems overall to show a muted level of enthusiasm for technology- or data-based products and services [83], compared with many countries in West or East Asia, suggesting that they have not been effectively engaged in communication about the benefits and imperatives of Smart Cities and related developments.

The reasons for this are currently open to speculation. Alsheddi et al. [84] discuss Saudi Arabia specifically and observe that "cultural factors, social factors, and religious factors are found to have an effect on the adoption of technological innovations in Saudi Arabia" (p. 53): this is in keeping with much of the current literature, including, for example, Mutambik et al. [9], who make similar reference to the Hofstede cultural dimensions but assign less prominence to the religious factors. According to Alsheddi et al. [84], these religious factors are deeply entangled with cultural factors and cannot be neglected. Clearly, the GCC countries are all majority Muslim and Islamic values are central to their peoples' responses to many things. It is important to consider whether and how this issue may bear on the development of approaches to OGD and Smart Cities [85].

There is a strand in the literature around OGD, and indeed Society 5.0, that emphasises its connection with open government and then the connection of that with certain freedoms of information and participation traditionally associated with "democratic" systems. This could lead to an argument that the "democratic deficit" identified in at least some GCC countries [84] may be holding back development. It has often been further suggested that Islamic values in themselves are somehow the source of this deficit. There could be a temptation to conclude that this is the basis for the observations of Alsheddi et al. and that religious factors may be inhibiting participation in technological and civic developments more broadly, and hence fundamentally undermining the Smart Cities project overall.

However, Sarkissian [86] argues strongly against this move, claiming that there has been confusion between factors that are religious and those that are mainly political. Sarkissian argues that it is government regulation of religion, rather than religious values as such, that has restricted participation. "In countries with high levels of religiosity but low levels of opportunity for political or civic involvement, participation in religious organizations is often the only means for public participation"; and governments often restrict the variety or scope of religious organisations. Similarly, from our perspective, participation in technological, civic and cultural developments becomes filtered through processes that may have a religious component or dimension but are perhaps more political underneath.

We see this as tied in with our theme of communication. At present, the cultural emphasis on traditional values, as identified in the Hofstede profile, is confusingly enmeshed with religious values and with political policies. Broader public participation may be held back by these factors. Where a cultural profile includes a strong tendency for people to look to leadership for direction, it is especially important for that leadership to be clear. Vagueness or equivocation in leadership leads rapidly to paralysis [87]. The government may set new objectives without changing existing objectives or constraints that contradict the new ones. The tendency towards caution and avoidance of risk acts against innovation unless there is great clarity that innovation will be rewarded and systems of penalties have been changed. People need to be explicitly encouraged, indeed, to be "given permission" to engage more fully. This includes articulating the objectives, the policies and the implications on a variety of cultural levels and recognising that the new processes enabled by Smart City developments will offer new and different methods of public participation that need to be embraced rather than viewed with suspicion.

Our more recent group of interviewees clearly recognise these directions.

> [T]here is unrealized potential to leverage data to connect people to their surroundings and give them a voice in how their cities develop. (PP5)

> Data can be used to build civil commons in several ways. For example, data can be used to create new audiences around shared interests, visualize otherwise abstract problems

*such as air pollution, and thus make them actionable, create compelling narratives about community interests and ideals (what kind of city do we want?), and pool resources to support group efforts and contribute to the commons.*
*(PP5)*

*We try to determine how the individual contributes to the solution, how he responds, and how he participates in the design and delivery of certain services. This, in our opinion, is a crucial element in smart cities. (PP1)*

*Currently, the city offers open data supported by commercial applications, which is great, because it facilitates communication between citizens and the city. It serves as a platform for residents to express fears or difficulties . . . (PP1)*

*The transformation into a smart city is complex and multidimensional, . . . It is based on six basic pillars: smart economy (competitiveness), smart people (social and human capital), smart governance (participation), smart mobility (transportation and information and communication technology), smart environment (natural resources) and smart living (quality of life). (PP3)*

We have seen that these goals and imperatives are not always appreciated or shared by people at different levels of organisations or the general public. There are a variety of reasons for this, but all can be mitigated by clearer communication, especially from the governmental level, which needs to explain the kinds of changes in attitudes and actions that are recognised as necessary at all levels.

## 6. Conclusions

Overall, it is clear from the results of this study that many questions remain to be answered before an unambiguous and certain path to smart cities/Society 5.0 can be defined. One of these questions, and a topic for future research, is the extent to which such a goal, which implies smart and open government, is ideologically consistent with the aims of GCC Counties and their leaderships. There is a serious issue here since, as suggested earlier, smart cities are a particularly important type of development for GCC countries. Not only is increasing urbanisation being embraced and encouraged but the concepts of smart cities are embedded foundationally in futuristic developments such as Saudi Arabia's "Neom" [88]. The ultimate success of developments like these will depend on success along all the dimensions of a smart city: Ojo et al. [89] identify several ways of classifying these, but while proposed dimensions differ in detail across the literature, they all share an emphasis on aspects that are essentially social and bound tightly with governance. A recognition of the importance of transparency and participation is pervasive [7,90]. There is a general recognition that achieving the potential of society is not just a matter of technologies and infrastructures but is fundamentally dependent on large-scale social change and development, which may also entail political change.

Some in, for example, Oman are urging the importance of open data and smart city concepts "to lay down the foundations for an empowered knowledge-based society" [43]. It is clear that GCC countries must act to enable these developments [91] and also that they need to act swiftly since the social as well as technical bases of transparency and participation need to be "designed in" at an early stage [92].

However, even without an accepted roadmap in this direction, this study has shown that there are several real short- and medium-term actions that could be taken to accelerate the progress of OGD initiatives, which are mission-critical to the ultimate aim of becoming a smart city. These actions include the removal of risks that staff attach to assuming a leadership role; these risks are currently a major barrier for many able and qualified individuals who would make effective and capable leaders. A particularly key action is the implementation of communication and education programmes designed to promote the benefits of smart government/cities and OGD to all staff, particularly potential leaders. It is imperative for such programmes to articulate to citizens at all levels the need for increased participation and engagement. There is a need to address and clarify possible conflicts

with traditional and/or religious values and practices, developing policies that allow people to understand expectations and limitations much more helpfully for governance and facilitation of the uses of technologies.

However, while this study has provided important insights into the barriers to OGD progress in GCC Countries, further research on a number of issues would prove valuable. A more holistic understanding of practice and policy within GCC Countries, for example, would be extremely useful. This could be gained by comparing the OGD performance of GCC Countries against the performance of more experienced OGD nations. It would also be useful to gain an insight into the typical profile of OGD users within wider society, both within and beyond the GCC Countries. This would provide guidelines for establishing more effective demand-side drivers, such as ease of use and the availability of appropriate datasets. This, in turn, would increase public use of OGD and help to generate a virtuous circle of confidence between governments and their public. While all this is not sufficient on its own to increase progress toward Society 5.0, it is a vital first step.

**Author Contributions:** Conceptualization, I.M., J.L. and A.A.; methodology, I.M. and A.A; validation, I.M., J.L., A.A. and J.Z.Z.; formal analysis, A.A. and J.Z.Z.; writing—original draft preparation, I.M., J.L. and A.A.; writing—review and editing, I.M., J.L., A.A. and J.Z.Z. All authors have read and agreed to the published version of the manuscript.

**Funding:** This research was funded by the Researchers Supporting Project number (RSP2023R233), King Saud University, Riyadh, Saudi Arabia.

**Institutional Review Board Statement:** The study was conducted according to the guidelines of the Declaration of Helsinki and approved by the Institutional Review Board (Human and Social Researches) of King Saud University.

**Informed Consent Statement:** Informed consent was obtained from all subjects involved in the study.

**Data Availability Statement:** Data available on request due to restrictions of privacy.

**Acknowledgments:** This research was funded by the Researchers Supporting Project number (RSP2023R233), King Saud University, Riyadh, Saudi Arabia.

**Conflicts of Interest:** The authors declare no conflict of interest.

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
