# Peer review of "Transitioning to Smart Cities in Gulf Cooperation Council Countries: The Role of Leadership and Organisational Culture"

_sustainability, doi:10.3390/su151310490_

Round 1
Reviewer 1 Report
The assessed scientific study in the recommended scope deals with a topic that I consider to be very current and interesting. Considering the specifics of the issue of Society 5.0 in Japan, it is always, in my opinion, a topic suitable for scientific treatment in the form of not only a study. Constantly existing and troubling human society, problems related to several factors in the field of urban development are and will be a suitable object of investigation for a long time. The reason is that this issue is open to new proposals, procedures and recommendations. Based on these facts, there is an obvious premise for scientific research, which the authors explain in the Introduction as well as in the second chapter under the title "Literature Review".
Given the nature of this scientific work, one of the key parts of its processing is the appropriate choice of a methodological approach to solving the investigated issue. The authors address this question very responsibly in the third chapter "Research methods". Considering the nature of the scientific study, the authors correctly chose and used selected scientific research methods. The result of their use is several new findings, which is the main goal of every scientific study.
In my opinion, the basic prerequisite for the high-quality processing of scientific work is a thorough preparation consisting in gathering sources of knowledge from which the authors want to draw ideas and knowledge. The list of used literature (References) shows that the authors have collected a sufficient amount of literature (70) scientific sources and other sources of knowledge that they used in the assessed scientific work. I am glad that they also included in this list such scientific sources, the content of which objectively represents the center of gravity of the scientific study, without which it would not be possible to even think about processing this topic. I appreciate that, despite the specificity of the topic, the authors used the knowledge of various foreign literature as well as researched in the globally recognized scientific databases Web of Science and SCOPUS, which contain valuable scientific works.
The content and formal page of the assessed scientific study fully meets the requirements of the MDPI publishing house published on the journal's website. It is appropriately and proportionally divided, the individual chapters logically follow each other and create one homogeneous whole.
The authors rightly devote the largest space to the "Results" part, which represents the focal part of the work. From the content of this part, the authors, based on their scientific research, present really new knowledge in chapters 5 and 6 on pages 11 to 17. In this way, they clearly fulfill the set goals of the scientific study. Especially in the final sixth chapter, the main contribution of the scientific study is expressed, where the authors propose their own recommendations and measures to eliminate obstacles to progress in building smart cities.
Author Response
Dear Reviewer,
We very much appreciate the time and effort you have given to read on our manuscript, and thank you for your kind and supportive comments. These are warmly appreciated.
Sincerely yours,
Ibrahim Mutambik,
Reviewer 2 Report
Please see my comments as below:
1. The study analyses the idea of Society 5.0, launched by Japan in 2016, as it is becoming more widely recognised as a blueprint for the growth of social infrastructure. It looks at how organisational culture and leadership affect projects using open government data (OGD) and the transition to smart cities in Gulf Cooperation Council (GCC) nations. Data from earlier semi-structured interviews with senior government employees in OGD-related roles is re-analyzed in this study. Instead of analysing external elements like technology and resources, the analysis focuses on internal aspects like leadership attitudes and organisational culture. The results show a leadership gap that impedes the growth of smart cities, particularly in the communication of policy, objectives, and strategies.
2) While the topic is interesting, the study can be improved as follows:
a) Proof reading will benefit the paper. There are several typos and inconsistencies throughout the paper. For instance the use of "study" and "paper"
2) It is not clear whether OGD is important to "smart cities" or is it vice-versa
3) A comparison with other jurisdictions is missing and is warranted.
4) The motivation and contribution of the paper are not clear. While it is obvious why OGD is important specific examples need to be listed especially in the context of GCC nations.
5) GCC governments are known to be strict with censoring of information flows. How will this impact OGG in these countries? Will it be a selective flow? If yes, then it can negate the basic principles of OGD.
6) There needs to be more details provided regarding the interview questions that were asked.
7) You only sampled 24 entities across GCC nations. Is it a sufficient sample size? Needs more discussion in this area. For instance, how did the sample break down by the GCC nations. What is the percentage of state entities etc? The concern is that the sample might not be representative enough.
Overall the paper is well written. A comparison with other jurisdictions can help improve the paper.
Thanks
Some improvement in grammar is needed.
Author Response
Dear Reviewer,
We very much appreciate the time and effort you have given to commenting on our manuscript. The points you have raised are very helpful. We have carefully reviewed the comments and revised the manuscript accordingly. Our responses are given in the attached file.
We have also submitted a revised version of our manuscript, and all the changes are marked with track changes.
We look forward to your reply on our revisions. We are happy to make any further changes that will improve the paper. Thank you again for your attention and consideration.
Sincerely yours,
Ibrahim Mutambik,

Reviewer 3 Report
Dear Authors
I have carefully read your manuscript entitled Transition to smart cities in Gulf Cooperation Council Countries: the role of leadership and organizational culture along with its previous version, i.e., Open Government Data in Gulf Cooperation Council Countries: An Analysis of Progress published with Sustainability in 2022.
The version sent for review reads very well, better than the 2022 publication, which increases the chances of reaching a wider range of readers representing different disciplines. All the more attention should be paid to carefully setting the presented material in a broader context, especially since the results of your research have already been published in the same scientific journal.
1. You declare an analysis of the impact of a specific leadership model and a specific organizational culture on the implementation of OGD. However, you present the culture itself either in an extremely cursory manner and with the help of little-inputting generalities (712-715) or leaving the reader in the realm of conjecture (609-610, 620-621). Consequently, the entire Section 5 could have been written independently of your research, and vice versa - your results can be applied to any model of administration reaching similar (and similarly general) conclusions.
2. The chosen method has numerous limitations, as you point out yourself, but you make no attempt to mitigate these limitations — from presenting a profile of the participants in more detail than the laconic "senior civil servants" to making more extensive excerpts from the interviews available as an appendix. How were the participants selected? What departments do they represent? What is their position in the hierarchy of decision-making processes? And so and so forth.
3. The thematic groups appear to have been cut to fit the thesis (630-634), rather than having emerged through analysis. Thus, for example, the division into Human resources and Leadership seems artificial — both are in fact broadly Human factor — seems but arbitrary.
4. The discussion is simultaneously an introduction, development, and conclusion. Presenting another part of the research in it does not obscure the fact that the conclusions of both stages are not de facto discussed. The relationships that the article was supposed to analyze are presented very generally and in axiomatic terms (712-755). Ultimately, we do not know what elements of this specific leadership and organizational culture hinder the implementation of Society 5.0 and OGD and in what aspect.
5. The summary should not include references to subsequent publications or open further discussion threads.
If your goal is to examine how a specific leadership and organizational culture impacts the progress of OGD initiatives (18-20, 78-84), zoom in on that culture and, above all, juxtapose the voices of the two research groups. Then, in discussion, you will be able to seek an answer to the question you posed. Alternatively, change the assumptions of the text — the mentioned culture can be a background devoid of an outcome quantifier.
I hope you will find the comments above helpful in providing the final version of your paper.
Best regards.
Author Response

(The authors gave the same response as above.)

Round 2
Reviewer 3 Report
Dear Authors,
I appreciate your efforts and I have no further comments.
Best regards.
Author Response

(The authors gave the same response as above.)
